# The Expression and Activation of the NF-κB Pathway Correlate with Methotrexate Resistance and Cell Proliferation in Acute Lymphoblastic Leukemia

**DOI:** 10.3390/genes14101880

**Published:** 2023-09-27

**Authors:** Rafael Renatino Canevarolo, Nathalia Moreno Cury, José Andrés Yunes

**Affiliations:** 1Centro de Pesquisa Boldrini, Centro Infantil Boldrini, Campinas 13083-210, SP, Brazil; rafael.renatinocanevarolo@moffitt.org (R.R.C.);; 2Medical Genetics Department, Faculty of Medical Sciences, State University of Campinas, Campinas 13083-970, SP, Brazil

**Keywords:** acute lymphoblastic leukemia, methotrexate, nuclear factor kappa B, tumor necrosis factor alpha, drug resistance

## Abstract

Acute lymphoblastic leukemia (ALL) is the most common childhood cancer. Although its prognosis continually improves with time, a significant proportion of patients still relapse from the disease because of the leukemia’s resistance to therapy. Methotrexate (MTX), a folic-acid antagonist, is a chemotherapy agent commonly used against ALL and as an immune-system suppressant for rheumatoid arthritis that presents multiple and complex mechanisms of action and resistance. Previous studies have shown that MTX modulates the nuclear factor kappa B (NF-κB) pathway, an important family of transcription factors involved in inflammation, immunity, cell survival, and proliferation which are frequently hyperactivated in ALL. Using a gene set enrichment analysis of publicly available gene expression data from 161 newly diagnosed pediatric ALL patients, we found the *Tumor necrosis factor α* (TNF-α) *signaling pathway* via *NF-κB* to be the most enriched Cancer Hallmark in MTX-poor-responder patients. A transcriptomic analysis using a panel of ALL cell lines (six B-cell precursor acute lymphoblastic leukemia and seven T-cell acute lymphoblastic leukemia) also identified the same pathway as differentially enriched among MTX-resistant cell lines, as well as in slowly dividing cells. To better understand the crosstalk between NF-κB activity and MTX resistance, we genetically modified the cell lines to express luciferase under an NF-κB-binding-site promoter. We observed that the fold change in NF-κB activity triggered by TNF-α (but not MTX) treatment correlated with MTX resistance and proliferation across the lines. At the individual gene level, *NFKB1* expression was directly associated with a poorer clinical response to MTX and with both an increased TNF-α-triggered NF-κB activation and MTX resistance in the cell lines. Despite these results, the pharmacological inhibition (using BAY 11-7082 and parthenolide) or stimulation (using exogenous TNF-α supplementation) of the NF-κB pathway did not alter the MTX resistance of the cell lines significantly, evidencing a complex interplay between MTX and NF-κB in ALL.

## 1. Introduction

Methotrexate (MTX), a folic-acid antagonist, has been a cornerstone agent in the treatment of acute lymphoblastic leukemia (ALL) for over 70 years. As an antimetabolite, MTX impacts cellular metabolism in multiple and complex fashions. Despite its widespread use in clinics, MTX’s mechanisms of action, however, seem to not be fully understood.

High doses of MTX are used in cancer treatment, in which MTX’s inhibition of dihydrofolate reductase prevents the sequential conversion of folic acid into dihydrofolate and tetrahydrofolate, halting the synthesis of nucleotides and the methionine–homocysteine cycle—the main source of methyl radicals used by the cell—resulting in cell death. In autoimmune diseases such as rheumatoid arthritis, a low dose of MTX inhibits the bifunctional purine biosynthesis protein PURH (ATIC), which blocks the conversion of aminoimidazole carboxamide ribonucleotide into inosine monophosphate, thus ceasing purine and adenosine synthesis, the latter of which is a potent proinflammatory metabolite. In this context, MTX’s anti-inflammatory effect was shown to be exerted through the decrease in the activity of nuclear factor kappa B (NF-κB) via increasing the levels of lincRNA-p21 through a DNA-dependent protein–kinase catalytic subunit mechanism [1], an effect that was abrogated by folinic acid supplementation [2]. MTX was also shown to inhibit the degradation of the NF-κB repressor IkB-α [3], and reduced levels of phosphorylated RelA (p65) were found among rheumatoid arthritis patients receiving low-dose MTX therapy [2].

The NF-κB family is an important family of transcription factors involved in inflammation, immunity, cell survival, and proliferation. It responds to stimuli such as stress, cytokines, free radicals, ultraviolet radiation, low-density lipoprotein oxidation, and antigens [4,5,6]. For instance, basal NF-κB activity is required for hematopoietic stem cells’ self-renewal and differentiation into myeloid and lymphoid lineages [7,8,9]. An aberrant activation of this transcription factor is frequently encountered in cancer [10], including hematological malignancies [11]. In fact, tumor necrosis factor α (TNF-α), the cytokine that triggers NF-κB signaling, is involved in all stages of leukemogenesis, from cell transformation, proliferation, angiogenesis, and extramedullary infiltration to its participation in the tumor microenvironment and in leukemic cells’ immune evasion, survival, and resistance to chemotherapy [12]. Constitutive NF-κB signaling has been detected in 40% of acute myeloid leukemia cases, and its aberrant activity enables leukemia cells to evade apoptosis and stimulate proliferation [13].

Although recent genomic studies did not highlight mutations in NF-κB genes [14], these transcription factors seem to be constitutively activated in ALL [15,16,17]. Importantly, the NF-κB pathway may be implicated in therapy resistance as well. For instance, the NF-κB subunit RelA (p65) was an independent prognostic marker of survival with the capacity to predict the duration of response to therapy in chronic lymphocytic leukemia [18]. In childhood ALL, the upregulation of the NF-κB signaling pathway has been associated with poor outcomes [19,20,21]. High levels of TNF-α were found in the plasma of ALL patients [22], associating these values with higher white blood cell (WBC) counts at diagnostics [23] and worse outcomes in childhood ALL [24].

By analyzing publicly available gene expression data from a cohort of 161 pediatric ALL patients [25], we confirmed the association between the overexpression of the NF-κB signaling pathway and WBC count after *upfront* in vivo MTX treatment, suggesting a link between the NF-κB pathway, drug resistance, and, ultimately, disease outcome. We investigated the transcriptional state of the NF-κB pathway in a panel of ALL cell lines, corroborating in vitro the findings on patients’ gene expression data, and generated stable transduced ALL cell lines expressing an NF-κB reporter construct that incorporated a copy of the firefly luciferase gene containing an NF-κB binding site within its promoter region. As a result, the synthesis of luciferase (and subsequent luminescence) was directly proportional to the level of active NF-κB present within the cell, allowing for an association between gene expression and NF-κB activity. Lastly, pharmacological modulation of the NF-κB pathway was also performed with the aim of understanding the relation between the modulation of NF-κB activity and MTX resistance.

## 2. Materials and Methods

### 2.1. Cell Lines

Six B-cell precursor (Nalm6, Nalm16, Nalm30, REH, RS4;11, and 697) and seven T-cell ALL (T-ALL) cell lines (ALL-SIL, CCRF-CEM, HPB-ALL, Jurkat, Molt-4, P12-ICHIKAWA, and T-ALL-1) were studied. The cells were cultured in RPMI-1640 culture medium (Cultilab, Campinas, Brazil) supplemented with 10% fetal bovine serum (Cultilab), 100 IU/mL of penicillin, and 100 μg/mL of streptomycin (Sigma-Aldrich, St. Louis, MO, USA) and were maintained at 37 °C in a 5% CO_2_ atmosphere in all experiments.

### 2.2. Reagents

MTX and all other compounds were purchased from Sigma-Aldrich. Human recombinant TNF-α was purchased from PeproTech. All reagents were diluted according to the manufacturer’s instructions, and the stock solutions were stored at −20 °C. Dilutions of the compounds to be used in the experiments were prepared only at the time of use.

### 2.3. Primary ALL Gene Expression and Gene Set Enrichment Analysis (GSEA)

The gene expression data analyzed in this work were taken from the work of Sorich et al. [25] (Gene Expression Omnibus accession number GSE10255). Sample hybridization was performed on HG-U133A Arrays (Affymetrix, Santa Clara, CA, USA). Expression values were analyzed using a Transcriptome Analysis Console (Thermo Fisher Scientific, Waltham, MA, USA) via the Probe Logarithmic Intensity Error Estimation (PLIER) PM-MM method. A gene set enrichment analysis (GSEA) [26] was performed using GSEA software version 4.3.2 (https://www.gsea-msigdb.org/gsea/index.jsp, accessed on 4 August 2023), using the ∆WBC count on day 3 (please see Sorich et al. [25] for more details about how the ∆WBC was calculated) as a continuous phenotype and using Pearson’s metric for ranking genes.

### 2.4. Cell Lines’ Gene Expression and Single-Sample GSEA (ssGSEA) Analysis

Cell lines (5–10 × 10^6^ cells) in the exponential growth phase were maintained overnight in a fresh culture medium, and they then had their RNA extracted via the Illustra RNAspin Mini Isolation Kit (GE Healthcare UK Limited Little Chalfont, Buckinghamshire, UK). the samples were processed using the One-Cycle Target Labeling and Control Reagents Kit (Affymetrix) and hybridized on HG-U133 Plus 2.0 Arrays (Affymetrix). Expression values were obtained with the iterPLIER + 16 algorithm and expressed on a log2 scale. For additional details, please see Silveira et al. [27] A single-sample GSEA (ssGSEA [26]) of Cancer Hallmarks was conducted using the ssgsea-gui.r script from the SSGSEA 2.0 library (github.com/broadinstitute/ssGSEA2.0, accessed on 19 June 2023). The heatmap for the visualization of the Cancer Hallmarks’ normalized enrichment scores was created using the heatmap.2 function from the gplots library in R (version 4.3.0); hierarchical clustering was performed using the *complete* method, and the Euclidean distance was used to compute the distances between the rows of the matrix.

### 2.5. Cell Viability Assays

In total, 80 microliters of a cell suspension (4 × 10^5^ cells/mL) were seeded in each well in a 96-well cell culture plate, followed by 20 µL of a culture medium containing MTX or another drug. Negative controls received only the vehicle. Each dose was tested in 3 biological replicates. Experiments with TNF-α supplementation had the cytokine added once a day over the course of 4 days. After the desired treatment period, the conditioned medium in the wells was replaced with 0.2 mL of phosphate buffer (PBS 1X) containing 2 μmol/L of calcein AM (Sigma-Aldrich). The culture plates rested for 30 min before their fluorescence was read (excitation/emission: 492/518 nm). To account for the heterogeneous cell distribution, the fluorescence of each well was measured at 25 points (a 5 × 5 scan matrix), and the value was integrated (summed) afterwards. Alternatively, the MTT (thiazolyl blue tetrazolium bromide, Sigma-Aldrich) reduction test was used to determine cell viability. Once the treatment period was completed, 20 μL of MTT (5 mg/mL) was added to each well, followed by 4 h of incubation. The formazan crystals produced were dissolved by adding 0.1 mL of a solution of 10 mM HCl and 10% dodecyl sulfate. After incubating overnight, their absorbance was measured at 570 nm. Survival at each dose was determined in relation to a negative control. The use of calcein AM or MTT is indicated in each case.

### 2.6. Determination of the Doubling Time

Two hundred microliters of a 2.5 × 10^5^ cell/mL suspension was seeded per well in triplicate in a 96-well culture plate. Fifteen microliters of the cell suspension was collected daily to count the cells under a microscope (via exclusion using trypan blue). The doubling time of each cell line was determined from the proliferation curves obtained.

### 2.7. Transduction with NF-κB Reporter Vector

One hundred and forty microliters of a cell suspension (2.15 × 10^5^ cells/mL) was seeded in a 96-well culture plate. Ten microliters of a commercial viral suspension (Cignal Lenti NFκB Reporter [luc] Kit, CLS-013L, QIAGEN) was added into each well with polybrene (8 μL/mL), resulting in a multiplicity of infection of around 11 viral particles/cell. Transduction controls in which the commercial viral suspension was replaced by a culture medium were also prepared. The cells were subjected to *spin infection* for 60 min at 580× *g* and 37 °C, followed by 24 h of incubation. One hundred and twenty microliters of the conditioned medium was replaced with fresh culture medium, followed by another incubation for 48 h. Puromycin (1.5 μL/mL) was then added to the culture medium for the selection of the transduced cells.

### 2.8. Luminescence Assay with Transduced Cells

In total, 1.5 × 10^5^ cells were suspended in 100 μL of culture medium and seeded in a 96-well culture plate in the presence of either MTX (100 nmol/L), TNF-α (100 ng/mL) or a vehicle. Following incubation, the plates were centrifuged for 15 min at 400× *g*, and the supernatant was discarded via inversion. Seventy-five microliters of phosphate-buffered saline (PBS 1X) and the same volume of a luciferin solution (Dual Glo Luciferase, Promega) were added to each well. After 10 min of incubation, luminescence was measured using a Synergy H1 Hybrid Reader (BioTek Instruments, Winooski, VT, USA). The luminescence of each well was measured at 25 points (a 5 × 5 scan matrix with acquisition gain at maximum) with the values integrated (summed) afterwards.

### 2.9. Statistical Analysis

Using Prism (version 9, GraphPad software), *t*-tests, a Mann–Whitney analysis, estimates of IC_50_ values and proliferation rates, dose–response curves, and correlation plots were made. For the cell lines’ data, the Spearman correlation coefficient was preferred instead of Pearson’s given its nonparametric nature. *p* values ≤ 0.05 were considered significant.

## 3. Results

### 3.1. The NF-κB Signaling Pathway Is Overexpressed in MTX-Resistant Primary ALL

We investigated the clinical relevance of the NF-κB pathway in the context of MTX resistance by analyzing the publicly available gene expression data of a cohort of 161 therapy-naïve, newly diagnosed pediatric cases of ALL which were published by Sorich et al. [25]. In their study, the authors demonstrated that a high WBC count following 3 days of single-agent frontline therapy with MTX was predictive of shorter long-term disease-free survival. Although the study found no NF-κB family gene among the top 50 probe-sets associated with the WBC at day 3, nor was *NF-κB signaling* listed among the topmost enriched pathways associated with this clinical phenotype, determined via GenMAPP and GO-BP, we still wanted to investigate whether the NF-κB signaling pathway was associated with WBC following MTX treatment via a different algorithm. For this purpose, we performed a GSEA [26], checking for the enrichment of the 50 Cancer Hallmarks. We found that *TNF-α signaling* via *NF-κB* was the topmost enriched Hallmark associated with a high post-MTX WBC count (and was thus linked to a bad prognosis) (Figure 1a,b). We performed the same analysis only for the top (*n* = 40) and bottom (*n* = 40) quartiles of the patients, identified as *poor* and *good* responders to MTX, respectively; again, *TNF-α signaling* via *NF-κB* was significantly enriched among the *poor* responders (Appendix A).

In mammals, the NF-κB transcription factor family consists of five proteins, p65 (RelA), RelB, c-Rel, p105/p50 (NF-κB1), and p100/52 (NF-κB2), which associate with each other to form distinct transcriptionally active homo- and heterodimeric complexes [28]. In the cohort of 161 patients, we observed that the expression levels of *NFK1*, *REL,* and *RELA* (median log_2_ values of 6.57, 5.73, and 7.06, respectively) were significantly higher than those of *NFKB2* and *RELB* (median log_2_ values of 4.48 and 4.18, respectively) (Appendix A). These findings suggest that *NFKB1*, *REL*, and *RELA* play preponderant roles in ALL biology. Next, we performed an unpaired *t*-test to investigate whether the NF-κB family genes (*NFKB1*, *NFKB2*, *REL*, *RELA*, and *RELB*) were differentially expressed in *poor* vs. *good* responders. We found that *NFKB1* and *REL* were overexpressed among *poor* responders compared to *good* responders (Figure 1c,d), whereas the other NF-κB family genes (*NFKB2*, *RELA*, and *RELB*) were not differentially expressed (Appendix A).

The unique DNA-binding properties of distinct NF-κB dimers may influence the selective regulation of NF-κB target genes [29,30]. Therefore, we conducted a correlation analysis of NF-κB family gene expression across the entire cohort. We observed a positive correlation between *NFKB1/REL*, *NFKB1/RELA*, *NFKB2/RELA*, *NFKB2/RELB*, and *REL/RELA* (Appendix A). Interestingly, when we performed the same analysis within the subgroups of *poor* and *good* responders to MTX, we noted distinct correlations. Among the *good* responders, only *NFKB1*/*RELA* showed a positive correlation, whereas in the *poor* responders, we found direct correlations between *NFKB1/REL*, *NFKB1/RELA*, *NFKB2/REL*, *NFKB2/RELA*, and *REL/RELA*, (Appendix A). These findings suggest that not only higher expression but also coordinated expression of NF-κB family genes may favor the formation of specific NF-κB dimers. In turn, these dimers could promote transcriptomic programs associated with MTX resistance.

### 3.2. Expression of the NF-κB Signaling Pathway Is Associated with MTX Resistance in a Panel of ALL Cell Lines

To investigate whether the same direct relation between the *TNF-α signaling pathway* via *NF-κB* and MTX resistance found in primary samples was also observed in vitro in a panel of ALL cell lines, we performed an ssGSEA [26] to identify the differentially expressed Cancer Hallmarks across the cell lines. Our analysis showed that while most of the Hallmarks were either over- or under-expressed uniformly among the cell lines, a small subset of Hallmarks presented differential enrichment scores across the lineages, including *TNF-α signaling* via *NF-κB* (Figure 2a), with the BCP-ALL cell lines exhibiting a higher ssGSEA NES compared to the T-ALL cell lines (Figure 2b).

In a previous study [29], we treated ALL cell lines with increasing doses of MTX for 48 or 96 h and determined their resistance to the drug (IC_50_ values in Appendix A). We also determined the cell proliferation rate (in terms of doubling time) by allowing the cells to proliferate freely in complete culture media and in the absence of any treatment. We observed a strong positive correlation between the doubling time and MTX resistance [29], with less proliferative lines exhibiting increased resistance to the drug at 48 h, thus indicating that cycle-related features influence MTX resistance in short treatment periods.

In this work, we also found that the NES of the *TNF-α signaling pathway* via the *NF-κB* Hallmark was positively correlated with the IC_50_ of MTX at 48 h (Figure 2c) and its cellular doubling time (Figure 2d), thus offering a transcriptional explanation for the differential proliferation rate and MTX resistance observed across the cell lines. We also searched for NF-κB family genes whose expression was associated with MTX resistance. Similar to patient samples, *NFKB1* expression was positively correlated to MTX resistance (Figure 2e); however, *NFKB2* expression, which was not correlated to MTX resistance in the primary samples, appeared as inversely correlated to the in vitro resistance to the antifolate in the cell lines (Figure 2f). Regarding the co-expression of NF-κB family genes, we found a negative correlation between *NFKB1/REL* and *REL/RELA* (Appendix A) which indicates a distinct pattern compared to that observed in the primary samples.

### 3.3. MTX Modulates NF-κB Activity in a Cell-Line-Dependent Fashion

To study the crosstalk between NF-κB activity and MTX resistance, we generated stable, transduced ALL cell lines expressing an NF-κB reporter construct. These cell lines were engineered to incorporate the firefly luciferase gene under the control of a basal promoter element (TATA box), joined to tandem repeats of NF-κB transcriptional-response elements. The luciferase gene carries a protein-destabilizing sequence to reduce basal, noninduced luciferase activity. As a result, the synthesis of luciferase (and subsequent luminescence) was directly proportional to the level of active NF-κB present within the cell.

The transduced cells were treated with either MTX or TNF-α (which was used as a positive control). The extent of NF-κB activation varied significantly among the different transduced cell lines upon treatment with MTX and TNF-α. In general, a 9 h treatment with MTX had no effect on the activity of NF-κB except for an increase observed in Jurkat, 697, and REH (Figure 3a) (plots for the modulation of NF-κB at 16 and 24 h are shown in Appendix A). We also determined the activation of NF-κB by MTX or TNF-α in relation to the control (ΔNF-κB). We found that 4 (out of 6) BCP-ALLs showed strong NF-κB activity (ΔNF-κB > 10) in response to TNF-α, whereas 3 (out of 6) T-ALLs and 1 BCP-ALL presented an intermediate response (5 < ΔNF-κB < 8) and 3 T-ALLs and 1 BCP-ALL had a weak response (ΔNF-κB < 3) to TNF-α stimulation (Figure 3b).

Interestingly, we found that the activation of ΔNF-κB by TNF-α at 9 h exhibited a positive association with MTX resistance (IC_50_ at 48 h) (Figure 3c), the doubling time (Figure 3d), and the NES for the *TNF-α signaling* via *NF-κB* pathway (Figure 3e), suggesting that NF-κB activity is mainly determined at the transcriptional level and could be influencing both cell proliferation and short-term MTX resistance. At the individual NF-κB gene level, the activation of ΔNF-κB by TNF-α was positively correlated with *NFKB1* expression (Figure 3f) but inversely correlated with *REL* expression (Figure 3g). A correlation matrix depicting Spearman’s rank correlation coefficients and *p*-values between doubling time, MTX resistance, the expression of NF-κB family genes, and the activation of ∆NF-κB by either MTX or TNF-α treatment can be found in Appendix A.

To assess the impact of NF-κB pathway inhibitors on the resistance of cell lines to MTX, we evaluated the effects of two pharmacological inhibitors: BAY 11-7082, an irreversible inhibitor of IKKα responsible for phosphorylating the NF-κB inhibitory protein IκBα; and parthenolide, which directly inhibits NF-κB and prevents its translocation to the nucleus [30]. Both compounds, however, did not enhance the cytotoxicity of MTX (Appendix A). We also treated the cell lines with increasing doses of MTX in the presence of TNF-α to investigate the effects of an increase in NF-κB signaling on MTX resistance. Interestingly, TNF-α induced MTX resistance only in the BCP-ALL cell line, REH; the cytokine was lethal to CCRF-CEM and 697 at the highest dose tested (100 ng/mL daily) and stimulated cell proliferation—which was not accompanied by an increase in MTX resistance—in the T-ALL cell lines Molt4, T-ALL-1, and ALL-SIL (Appendix A). Based on these findings, it appears that exogenously induced NF-κB signaling has a limited and cell-specific impact, if any, on MTX resistance in vitro.

## 4. Discussion

MTX exerts its therapeutic effect through diverse and complex cellular mechanisms. While some treatment protocols for ALL use MTX as a single agent, others administer the antifolate in combination with other chemotherapeutic drugs. Previous works associated NF-κB with a poor prognosis [19,21]; however, these studies were either based on smaller cohorts or on targeted approaches (RT-PCR and antibodies). A study by Cleaver et al. [20], which used a GSEA analysis on microarray data, identified the expression of the NF-κB pathway as predictive of relapse, although this association was not exploited in the context of drug resistance to any chemotherapeutic agent. By performing a GSEA on publicly available gene expression data from bone-marrow-derived blasts from 161 newly diagnosed pediatric ALL patients, we identified constitutive *TNF-α signaling* via *NF-κB* pathway as the most enriched Cancer Hallmark among patients with higher WBC counts following a single-agent-based frontline therapy utilizing MTX. This allowed us to unequivocally link NF-κB pathway overexpression to MTX clinical resistance. We also found that *NFKB1* and *REL* were overexpressed in *poor* responders to MTX compared to *good* responders to MTX. Interestingly, in *poor* responders, the expression of these two genes was directly correlated, suggesting that higher and coordinated expression of these genes may contribute to the formation of the NFKB1/REL dimer. This dimer could potentially play a crucial role as a transcription factor in the development of MTX resistance in patients. Functional validation studies are needed to test this hypothesis.

We validated the findings obtained from patient samples in a panel of 13 ALL cell lines. An ssGSEA analysis on the cell lines’ transcriptomic data revealed that *TNF-α signaling* via *NF-κB*—one of the few Cancer Hallmarks that presented differential enrichment scores across the cell lines overexpressed in BCP-ALL compared to T-ALL—was also directly associated with the activation of NF-κB by TNF-α, cell proliferation, and MTX resistance, confirming the crosstalk between the expression of the signaling pathway and the phenotypic traits. At the individual gene level, *NFKB1* was overexpressed (i) among poor-responder patients to MTX; (ii) in MTX-resistant cell lines; and (iii) in samples with higher levels of NF-κB activation by TNF-α, evidencing the important role of this NF-κB family gene in pathway regulation and the phenotypes associated with it, such as resistance to MTX. Conversely, *REL* was overexpressed in poor responders to MTX but inversely correlated with the activation of NF-κB by TNF-α, whereas *NFKB2* expression was negatively correlated with MTX resistance only in cell lines, indicating a more context-specific role for these genes. It is worth noting that unlike the data from primary samples, the expression levels of *NFKB1* and *REL* were inversely correlated in the cell lines. This suggests a distinct transcriptional regulation of these transcription factor family genes in cell lines compared to patient samples. Additionally, since *REL* expression was not correlated with MTX resistance, it is plausible that other NF-κB dimers may be more relevant in in vitro contexts.

Using a panel of stable, transduced ALL cell lines, we showed that MTX was able to modestly increase NF-κB activity in only a few of the cell lines tested. Previous works had shown that MTX inhibited NF-κB activation by increasing both adenosine release and the activation of the adenosine receptor A2a in rheumatoid arthritis [31] and that MTX decreased the TNF-α-mediated activation of NF-κB in Jurkat through the inhibition of IκBα phosphorylation and degradation [3], as well as by releasing adenosine, although the dose (10 µM) and length of treatment (60 min) varied significantly from our experimental conditions. It is plausible to hypothesize that longer periods of treatment—like those tested in our study—may trigger stress responses and signaling pathways more distinct than those described by other authors.

Our results also showed that even in those cases in which MTX increased NF-κB levels, the magnitude of the activation was never higher than that induced by TNF-α treatment (the only exception was P12-Ichikawa at 16 h). This finding suggests that MTX may modulate NF-κB signaling through distinct mechanisms or pathways compared to direct stimulation by TNF-α. The differential effects on the activation of NF-κB by these two stimuli highlight the complexity of NF-κB regulation and warrant a further investigation to elucidate the underlying mechanisms and their implications in the context of MTX resistance. We also found that the TNF-α-mediated activation of NF-κB was positively correlated to MTX resistance and the cell-doubling time. There is a vast body of literature linking NF-κB signaling to cell proliferation which, in turn, is associated with cell susceptibility to short-term resistance MTX, thus making it reasonable that these parameters were correlated to one another.

Functionally, treatment with NF-κB inhibitors (BAY 11-7082 and parthenolide) did not sensitize the cells to MTX, nor did TNF-α supplementation induce resistance to the antifolate in the cell lines (except for REH). These pharmacological inhibitors target NF-κB in a nonspecific manner, suggesting that more specific NF-κB inhibitors may be required to elicit a more conspicuous additive/synergistic effect. However, we cannot rule out the possibility that the overexpression of the NF-κB pathway is associated with but does not directly promote MTX resistance in ALL cell lines; in this sense, the activation of NF-κB would be the byproduct of a transcriptional state characterized by MTX resistance but not the leading cause of it. In fact, the cellular processes attributed to TNF-α signaling via NF-κB can range from proliferation to apoptosis, depending on the signal strength, the signaling molecules recruited, and the crosstalk with other pathways [32].

In summary, we observed a direct association between the overexpression of the NF-κB pathway and a poorer clinical response to MTX, as well as between the overexpression of NF-κB genes, the activation of NF-κB by TNF-α, MTX resistance, and reduced cell proliferation in a panel of ALL cell lines. Further studies must be conducted to investigate whether inhibiting NF-κB (both the family genes and pathway) results in drug re-sensitization and therapeutic benefit in ALL.

## Figures and Tables

**Figure 1 genes-14-01880-f001:**
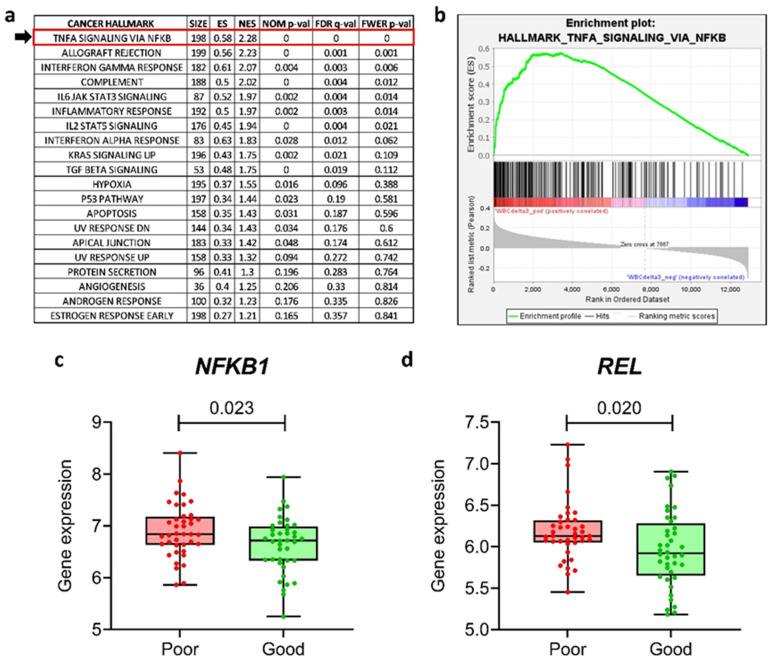
GSEA of 161 pediatric cases of ALL identified Cancer Hallmarks associated with higher circulating WBCs. (**a**) List of Hallmarks enriched in ALL blasts at diagnosis in samples with increased circulating WBCs (and thus associated with a worse prognosis) following MTX therapy. Highlighted is the topmost enriched pathway, *TNF-α signaling* via *NF-κB*, whose enrichment plot is shown in (**b**). (**c**,**d**) Expression levels of *NFKB1* and *REL* in the top (*n* = 40) and bottom (*n* = 40) quartiles with the poorest (*poor*) and best (*good*) clinical responses to MTX. *p*-values for the unpaired *t*-test. Analysis conducted on data from Sorich et al. [25] Abbreviations: ALL—acute lymphoblastic leukemia; ES—enrichment score; FDR q-val—false discovery rate *q*-value; FWER *p*-val—familywise error-rate *p*-value; GSEA—gene set enrichment analysis; MTX—methotrexate; NES—normalized enrichment score; Size—the number of genes that comprise the Hallmark; NOM *p*-val—nominal *p*-value; WBCs—white blood cells.

**Figure 2 genes-14-01880-f002:**
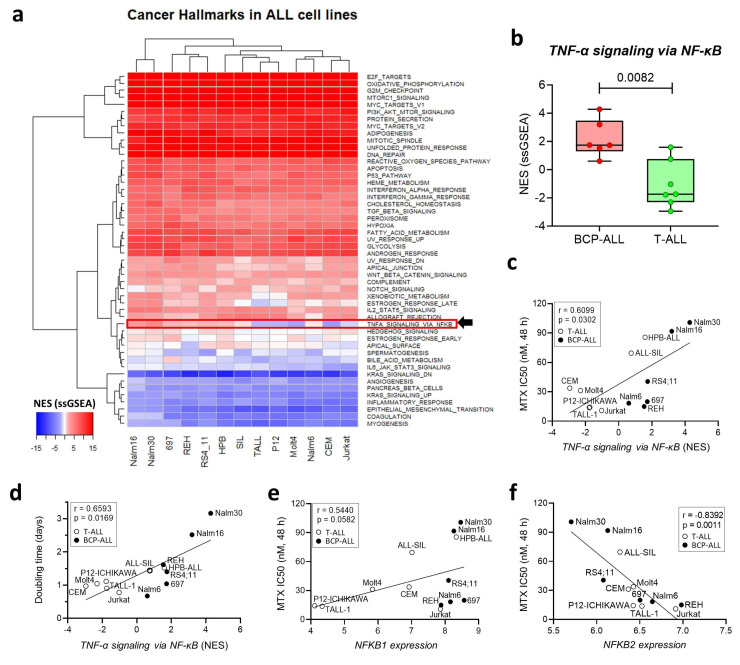
The transcriptional status of the NF-κB signaling pathway correlates with proliferation and MTX resistance. (**a**) The NES of the 50 Cancer Hallmarks per cell line, calculated via a ssGSEA. Highlighted is *TNF-α signaling* via *NF-κB*. (**b**) The NES of *TNF-α signaling* via *NF-κB* in the T-ALL and BCP-ALL cell lines (*p =* values for the Mann–Whitney test). (**c**) The association between *TNF-α signaling* via *NF-κB* NES and MTX resistance and (**d**) doubling time. (**e**) The correlation between *NFKB1* or (**f**) *NFKB2* expression and MTX resistance. r and *p* refer to Spearman’s correlation coefficient and the *p*-value, respectively. BCP-ALL and T-ALL are depicted as black and white dots, respectively. Abbreviations: ALL—acute lymphoblastic leukemia; BCP-ALL—B-cell precursor acute lymphoblastic leukemia; MTX—methotrexate; NES—normalized enrichment score; NF-κB—nuclear factor kappa B; ssGSEA—single-sample gene set enrichment analysis; T-ALL—T-cell acute lymphoblastic leukemia; TNF-α—tumor necrosis factor α.

**Figure 3 genes-14-01880-f003:**
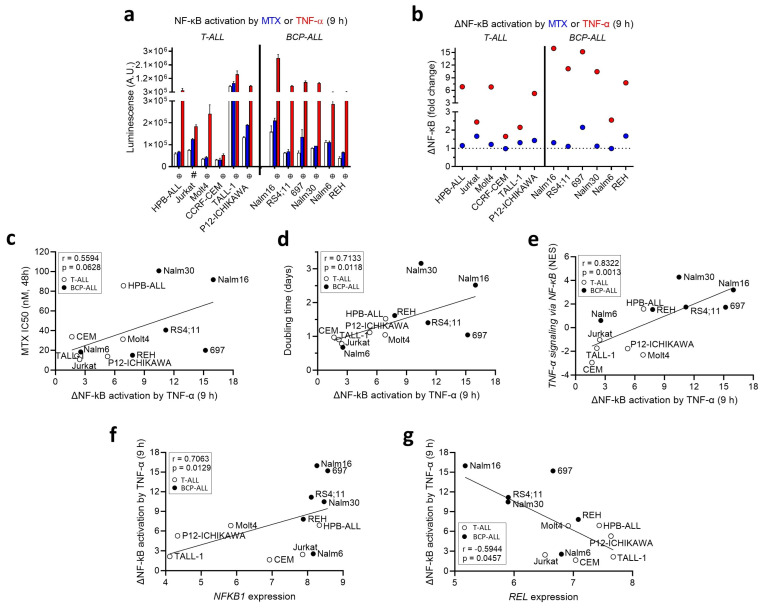
NF-κB activation induced by TNF-α correlates with proliferation and MTX resistance. (**a**) NF-κB signal in cell lines treated with MTX (100 nmol/L, in blue), TNF-α (100 ng/mL, in red) or a vehicle (Ctrl, white bars) for 9 h. A one-way analysis of variance followed by Tukey’s post test was performed for each cell line. ⊕: TNF-α differed from Ctrl and MTX, which did not differ between each other. #: All 3 groups differed from one another. (**b**) The ratio between the averages in (**a**), depicting the fold change (∆) in the modulation of NF-κB by MTX (blue dots) or TNF-α (red dots) in relation to the control (dashed line, normalized per cell line). (**c**–**g**) The association between ∆NF-κB activity triggered by TNF-α and MTX resistance, doubling time, the *TNF-α signaling* via *NF-κB* pathway, determined via the NES from the ssGSEA analysis, and *NFKB1* and *REL* expression, respectively. r and *p* refer to Spearman’s correlation coefficient and the *p*-value, respectively. BCP-ALL and T-ALL are depicted as black and white dots, respectively. Abbreviations: ALL—acute lymphoblastic leukemia; A.U.—arbitrary units; BCP-ALL—B-cell precursor acute lymphoblastic leukemia; MTX—methotrexate; NES—normalized enrichment score; NF-κB—nuclear factor kappa B; ssGSEA—single-sample gene set enrichment analysis; T-ALL—T-cell acute lymphoblastic leukemia; TNF-α—tumor necrosis factor α.

## Data Availability

The data presented in the study are deposited in the Gene Expression Omnibus repository under the accession number GSE218348.

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
