# Peer review of "The Expression and Activation of the NF-κB Pathway Correlate with Methotrexate Resistance and Cell Proliferation in Acute Lymphoblastic Leukemia"

_genes, 2023, doi:10.3390/genes14101880_

Round 1
Reviewer 1 Report
Comments on the manuscript "Expression and activation of the NF-κB pathway correlates with methotrexate resistance and cell proliferation in acute lymphoblastic leukemia" ID genes-2590797
As indicated in the title, in this article the relationship between methotrexate resistance and cell proliferation with the expression and activation of the NF-kB pathway by TNFa is demonstrated through an enrichment analysis of accessible data from ALL patients and its in vitro confirmation with B-lineage and T-lineage ALL cell lines. This addresses a relevant health problem, since MTX is one of the most widely used drugs in cancer treatment, and resistance to this drug compromises therapeutic success. The approach from the experimental point of view is well controlled, there is no doubt about the overall results that are reinforced by the intercellular variability of the different lines and the tests with different inhibitors.
The scope of the results is limited by dealing with cell lines that leave out variables that usually come into play in patients such as the pharmacogenetics of the molecules involved in the transport and metabolism of MTX. In fact, treatment of the cell lines with NF-kB inhibitors does not reverse MTX resistance, which suggests that NF-kB activation is a consequence of MTX resistance and not the cause itself, so the factors that may be mediating the correlation described remain to be explored in greater depth. In any case, the contribution is valuable and serves as a basis for future studies. I only detected what could be an easy-to-correct error. In the text, when referring to Figure 3c, the effect of TNF-a in association with resistance to MTX at 48 hours is mentioned; however, the figure shows a graph with the results at 9h.
Author Response
We appreciate and thank the reviewer for their comments.
Regarding the point raised, "9 h" refers to the time at which ΔNF-κB activation triggered by TNF-α was accessed (x axis of the plot in Figure 3c), while "48 h" refers to MTX exposure time for IC50 determination (y axis of the plot).
To avoid confusion, we have adapted the sentence in the manuscript: "Interestingly, we found that ΔNF-κB activation triggered by TNF-α at 9 hours exhibited a positive association with MTX resistance (IC50 at 48 h) (Figure 3c)..."
Reviewer 2 Report
In the manuscript, the authors identified the "TNF-α signaling pathway" as enriched in pediatric ALL patients and cell lines resistant to methotrexate, revealing a link between NF-κB activity triggered by TNF-α, MTX resistance, and proliferation; however, attempts to manipulate the NF-κB pathway did not significantly change MTX resistance, indicating a complex interaction. The manuscript is well written and organized. Below is one minor comment.
In Figures 2 (c-f) and 3 (c-g), the authors should label the black dots as “BCP-ALL” and white dots as “T-ALL” beside the line graph so that readers don’t have to check the legend for explanation.
Author Response
We appreciate and thank the reviewer for their comments.
We have labelled the black dots as “BCP-ALL” and white dots as “T-ALL” beside the line graph in Figures 2c-f and 3c-g, to facilitate the understanding of the plots.
Reviewer 3 Report
The article by Canevarolo R.R. et al.:” Expression and activation of the NF-κB pathway correlates with methotrexate resistance and cell proliferation in acute lymphoblastic leukemia” tries to correlate the drug resistance mechanism induced by Methotrexate (MTX) treatment in ALL cells. The authors correlate NF-kB activity triggered by TNF-alpha with MTX resistance and proliferation across the ALL-human cell lines suggested as a mechanism in resistant ALL cells. Two members of the NF-kB family, NF-kB1 and Rel, are primarily involved.
- The pharmacologic inhibition or exogenous activation of NF-kB is unable to alter MTX resistance. This sounds strange, but additional experiments need to be performed either with another inhibitor or by silencing specific monomers of the NF-kB transcription factor.
- Page 5. Do they have any evidence that the dimer NF-kB1/c-Rel is working in the context of MTX resistance in ALL? Or at least comment on it.
- Page 6. “… We observed a strong positive correlation (Spearman r=0.7483; p=0.0051) between doubling time and MTX resistance, with less proliferative lines exhibiting increased resistance to the drug at 48 h, thus indicating that cycle-related features influence MTX resistance in short treatment period…”. It is not clear to what figure are refereed these r and o p values.
- Page 7. The comments in Figures 2e and 2f seem not to correspond. Does the data on human samples partially differ from those of human cell lines? Please clarify this point.
Minor comments
Page 2, second paragraph. “…For instance, basal NF-κB activity is required for hematopoietic stem cells (HSCs) self-renewal and differentiation into myeloid and lymphoid lineages …” I suggest two more comprehensive papers: (T-cells) Adv Exp Med Biol. 2020;1227:145-164, (B-cells) Front Immunol. 2023 Jul 18;14:1214095.
Page 7. “ … Moreover, 4 (out of 6) BCP-ALLs showed a strong ΔNF-κB response …” Can you please explain Δ?
Page 7. “…These findings suggest that exogenously induced NF-κB signaling plays, if any, a very limited cell-specific role in MTX resistance in vitro…” Please check the sentence, it is not clear the meaning.
Moderate revision of English
Author Response
We appreciate and thank the reviewer for their comments. Please find our comments/answers below:
- Page 5. Do they have any evidence that the dimer NF-kB1/c-Rel is working in the context of MTX resistance in ALL? Or at least comment on it.
A: The work by Sorich et al. includes a supplementary table listing the 50 most differentially expressed genes between good and poor MTX responders. Notably, this list does not contain any NF-kB family genes. Additionally, the authors employed two algorithms for enrichment analysis: GenMAPP and GO-BP. It's worth noting that they did not utilize GSEA, which might explain why they did not identify enrichment of the "TNF-alpha signaling via NF-kB" pathway in poor responders. We have incorporated this information into the manuscript's text.
Additionally, in response to the reviewer's question, we conducted an investigation into the correlation between NF-kB family genes in the entire patient cohort, as well as within the subgroups of 'good' and 'poor' responders separately. Notably, NFKB1 and REL (as well as other pairs of NF-kB genes) showed a positive correlation in the whole cohort and among poor responders. However, this correlation was not observed among good responders, suggesting that the NFKB1/REL dimer may play a role in the context of MTX resistance. Supplementary tables have been added to illustrate the correlation between the expression of NF-kB family genes in the entire cohort, as well as among poor and good responders.
Interestingly, NFKB1 and REL exhibited an inverse correlation in the cell lines, indicating a different regulation of NF-kB family genes in these cells compared to patient samples. While the correlation coefficient and p-values for these genes in the cell lines were previously presented as supplementary data, we have now explicitly mentioned the inverse correlation between these genes in the Results section.
Moreover, we have incorporated additional sentences in the Discussion section, such as: "NFKB1 and REL were overexpressed in MTX poor responders compared to good responders. Interestingly, in poor responders, the expression of these two genes was directly correlated, suggesting that their higher and coordinated expression may contribute to the formation of the NFKB1/REL dimer. This dimer could potentially play a crucial role as a transcription factor in the development of MTX resistance in patients. Functional validation studies are needed to test this hypothesis.".
Page 6. “… We observed a strong positive correlation (Spearman r=0.7483; p=0.0051) between doubling time and MTX resistance, with less proliferative lines exhibiting increased resistance to the drug at 48 h, thus indicating that cycle-related features influence MTX resistance in short treatment period…”. It is not clear to what figure are refereed these r and o p values.
A: In this context, we would like to clarify that the association mentioned was observed and published in our previous work (Canevarolo et al., Frontiers in Oncology 2022). We included a reference to our published work in the sentence to make it clearer for the readers that this association was described in another publication. We have also removed the mention of the Spearman coefficient and P-values of the correlation, as this information is irrelevant in the context of this manuscript and can be misleading to the readers.
- Page 7. The comments in Figures 2e and 2f seem not to correspond. Does the data on human samples partially differ from those of human cell lines? Please clarify this point.
A: We have reformulated the phrase to improve its clarity: "Similar to patient samples, NFKB1 expression was positively correlated to MTX resistance (Figure 2e); NFKB2 expression, however, which was not correlated to MTX resistance in primary samples, showed up as inversely correlated to the in vitro resistance to the antifolate in the cell lines (Figure 2f)"
Minor comments
Page 2, second paragraph. “…For instance, basal NF-κB activity is required for hematopoietic stem cells (HSCs) self-renewal and differentiation into myeloid and lymphoid lineages …” I suggest two more comprehensive papers: (T-cells) Adv Exp Med Biol. 2020;1227:145-164, (B-cells) Front Immunol. 2023 Jul 18;14:1214095.
A: Done as suggested. We thank the reviewer for suggesting these references.
Page 7. “ … Moreover, 4 (out of 6) BCP-ALLs showed a strong ΔNF-κB response …” Can you please explain Δ?
A: "ΔNF-κB" represents the fold change in NF-kB activity induced by either TNF-alpha or MTX relative to the control. We employed ΔNF-κB for comparing NF-kB activity across different cell lines, each of which possesses its own constitutive NF-kB activity level. We apologize for any previous lack of clarity and have now clarified this in the text. The reformulated sentence reads as follows: "We also determined NF-κB activation by MTX or TNF-α in relation to control (ΔNF-κB). We found that 4 (out of 6) BCP-ALLs showed a strong NF-κB activity (ΔNF-κB > 10) in response to TNF-α; whereas 3 (out of 6) T-ALLs and 1 BCP-ALL presented an intermediate response (5 < ΔNF-κB < 8), and 3 T-ALLs and 1 BCP-ALL had a weak response (ΔNF-κB < 3) to TNF-α stimulation (Figure 3b)."
Page 7. “…These findings suggest that exogenously induced NF-κB signaling plays, if any, a very limited cell-specific role in MTX resistance in vitro…” Please check the sentence, it is not clear the meaning.
A: We reformulated the sentence to improve its clarity: "Based on these findings, it appears that exogenously induced NF-κB signaling has a limited and cell line-specific impact, if any, on MTX resistance in vitro".
We would also like to mention that the revised version of the manuscript has undergone professional English revision.